# Deep Active Learning in the Open World

**Tian Xie**                                                    *tian.xie@wisc.edu*
**Jifan Zhang**                                                 *jifan@cs.wisc.edu*
**Haoyue Bai**                                              *baihaoyue@cs.wisc.edu*
**Robert Nowak**                                              *rdnowak@wisc.edu*
*University of Wisconsin-Madison*

**Reviewed on OpenReview:** *https://openreview.net/forum?id=HkmymFPODz*

## Abstract

Machine learning models deployed in open-world scenarios often encounter unfamiliar conditions and perform poorly in unanticipated situations. As AI systems advance and find application in safety-critical domains, effectively handling out-of-distribution (OOD) data is crucial to building open-world learning systems. In this work, we introduce ALOE, a novel active learning algorithm for open-world environments designed to enhance model adaptation by incorporating new OOD classes via a two-stage approach. First, diversity sampling selects a representative set of examples, followed by OOD detection with GradNorm scores to prioritize likely unknown classes for annotation. This strategy accelerates class discovery and learning, even under constrained annotation budgets. Evaluations on three long-tailed image classification benchmarks demonstrate that ALOE outperforms traditional active learning baselines, effectively expanding known categories while balancing annotation cost. Our findings reveal a crucial tradeoff between enhancing known-class performance and discovering new classes, setting the stage for future advancements in open-world machine learning. Code of the work will be be uploaded at `https://github.com/EfficientTraining/LabelBench`.

## 1 Introduction

Modern machine learning models have achieved remarkable progress by leveraging large amounts of labeled data (LeCun et al., 2015; He et al., 2016; Khosla et al., 2020). Despite this success, most models are developed for closed-world settings, assuming that both training and test data originate from the same distribution. However, this assumption does not align with real-world environments, where models are inevitably encounter out-of-distribution (OOD) data with previously unseen classes (Hendrycks & Gimpel, 2022; Hendrycks et al., 2022; Salehi et al., 2022). Constrained by their fixed class boundaries, traditional models often struggle to generalize effectively to these novel classes, limiting their adaptability in open-world scenarios. Furthermore, obtaining human supervision for these novel examples in open-world scenarios is often time-consuming and costly, posing a significant challenge to model adaptation and improvement. Active learning, which iteratively selects the most informative examples for labeling, has emerged as a promising approach to address the expensive nature of gathering human supervision. By prioritizing examples that provide the most significant information gain, active learning can enhance the model's learning efficiency while reducing the need for extensive human annotation. This approach is particularly valuable in open-world scenarios, where the high annotation cost and time-consuming nature of labeling make traditional supervised learning methods impractical.

Despite its potential, existing active deep learning algorithms (see Zhan et al. (2022); Zhang et al. (2024a) for overviews) have rarely been studied under open-world scenarios, particularly those involving novel classes and imbalanced data distributions. In this work, we addresses this gap by developing a novel active learning algorithm that integrates OOD detection techniques with GradNorm scores to handle the complexity of open-world environments. Our method is designed for multi-class classification tasks in the open-world, where the model encounters both known and unknown classes after deployment. Of the few works that study

active learning under open-world settings, existing methods are often tailored to specific vision tasks, which result in highly specialized algorithm designs. When these methods are simplified for more generic problems like classification, they often reduce to basic uncertainty or diversity sampling techniques. In contrast, our approach in this paper offers a more comprehensive and versatile solution. By bridging OOD detection and diversity in our sampling strategy, we provide a more comprehensive sampling strategy compared to approaches that focus on only one of these aspects.

We propose **ALOE** (**A**ctive **L**earning in **O**pen-world **E**nvironments), a two-stage algorithm tailored for open-world active learning. ALOE addresses the unique challenges of class discovery in open-world settings by dynamically incorporating new OOD classes through a structured sampling strategy: In the first stage, ALOE performs diversity sampling to select a representative set of examples from the data pool. This sampling ensures broad coverage of the data distribution, capturing a range of potential new classes and concepts. By focusing on diversity, the algorithm increases the probability of identifying and learning from rare or infrequent classes that may otherwise be overlooked in a random sampling approach. The second stage leverages GradNorm scoring function to rank examples within each cluster, prioritizing instances most likely to belong to unknown OOD classes. The OOD detection with GradNorm scores provides a unified framework that distinguishes between in-distribution (ID) and OOD examples with greater resilience to model overconfidence. By focusing annotation efforts on these high-priority examples, ALOE accelerates the discovery and learning of new classes, even when the annotation budget is limited.

To empirically validate our approach, we conducted experiments on long-tail imbalanced image classification datasets. This choice of datasets is motivated by the prevalence of long-tail distributions in real-world scenarios, where rare classes or concepts are often underrepresented. Such imbalanced distributions make random sampling ineffective in discovering unknown classes, particularly those with small sample sizes. Our experimental results demonstrate the effectiveness of our approach. Compared with random sampling, on ImageNet-LT, ALOE saves 70% of annotation cost to achieve the same accuracy. In six out of six experimental settings (Sections 5.2.2 and 5.2.3), we observed that our algorithm performs the best comparing to all baseline experiments. These highlights underscore the potential of our method to significantly enhance model adaptation in open-world environments.

Lastly, our work reveals a novel tradeoff between improving performance on known classes and discovering new ones. This finding opens up an important avenue for future research, as balancing these competing objectives is crucial for developing truly adaptive AI systems. In summary, our proposed active learning algorithm offers a promising solution for adapting machine learning models to new, previously unseen conditions in open-world environments. By efficiently incorporating unknown instances and minimizing human annotation efforts, our approach paves the way for more robust and adaptable AI systems capable of adapting to dynamic, real-world scenarios.

## 2 Related Work

The earliest machine learning methods typically employed passive learning in a closed-world setting, where the model passively received training data and all data was fully labeled. However, in modern research and practice, it is common to encounter situations where *i)* some or all of the data is unlabeled, *ii)* the unlabeled data contains new classes, or *iii)* it is necessary to actively select training data. Our research combines these aspects and proposes a novel approach tailored to these scenarios.

### 2.1 Open-World Learning

Open-world learning addresses the challenge of operating in dynamic environments where the model starts with a set of known classes and must detect and manage instances from unknown classes. This paradigm involves both labeled and unlabeled data, with a mixture of known and unknown classes, requiring the model to either classify examples into known categories or recognize them as novel (Rizve et al., 2022; Zhu et al., 2024; Xiao et al., 2024). Unlike novel class discovery that focuses solely on discovering new object categories (Fini et al., 2021; Zhong et al., 2021b; Han et al.; Zhong et al., 2021a; Roy et al., 2022), open-world learning must also correctly identify instances from previously known classes, making the task more complex.

Open-world semi-supervised learning (Cao et al., 2022; Sun et al., 2024) generalizes semi-supervised learning by considering scenarios where the data includes both labeled and unlabeled instances from known and unknown classes. The model must learn to classify known classes and identify novel ones from unlabeled data. In contrast to traditional supervised settings, the ability to generalize to unseen classes is crucial. Open-set recognition shares similarities with open-world learning by allowing the model to reject novel instances during testing (Ge et al., 2017; Sun et al., 2020; Geng et al., 2020). However, open-world recognition extends this by requiring the model to incrementally learn and incorporate these novel classes into the set of known categories (Boult et al., 2019; Bendale & Boult, 2015). Open-world contrastive learning further enhances this by learning compact representation spaces that facilitate both the classification of known classes and the discovery of novel ones (Sun & Li, 2022).

Open-world learning represents a significant advancement due to its capacity for active instance selection. In this setting, the model actively selects instances from unknown classes for labeling, allowing it to continuously expand the set of known classes as new data is labeled. Earlier work by Bendale & Boult (2016) laid the foundation for open-world recognition, introducing the Nearest Non-Outlier (NNO) algorithm and establishing evaluation protocols for managing novel classes in a continual learning environment. More recently, a comprehensive review by Zhu et al. (2024) highlighted three core components for open-world systems: rejecting unknowns, discovering novel classes, and incrementally learning from them. These components form the basis of many modern approaches to open-world learning. Our research extends this by integrating open-world learning with long-tail distributions and active sample selection, enabling the model to not only handle class imbalance but also dynamically evolve as new classes emerge.

## 2.2 Out-of-Distribution Detection

Out-of-distribution (OOD) detection is a fundamental challenge for machine learning models deployed in open-world environments. It involves identifying whether inputs belong to unknown classes that the model has not encountered during training. The overconfidence of neural networks when handling out-of-distribution data was first revealed in Nguyen et al. (2015). Research in OOD detection has taken several main directions: post-hoc methods that devise scoring functions for detecting OOD inputs (Liu et al., 2020; Lee et al., 2018; Sun et al., 2022), training-time regularization methods that leverage additional auxiliary OOD datasets to address OOD detection (Hendrycks et al., 2018; Van Amersfoort et al., 2020; Katz-Samuels et al., 2022; Bai et al., 2023; 2024), and approaches exploring representation learning, such as exploring multiview contrastive losses (Khosla et al., 2020; Chen et al., 2020) for OOD detection (Winkens et al., 2020; Sehwag et al., 2021; Ming et al., 2022).

In particular, post-hoc methods address OOD detection by deriving test-time OOD scoring functions for a pre-trained classifier. These methods include maximum softmax probability (Hendrycks & Gimpel, 2016), distance-based scores (Lee et al., 2018; Sun et al., 2022), energy-based scores (Liu et al., 2020), activation rectification (Sun et al., 2021), ViM score (Wang et al., 2022a), gradient-based scores (Huang et al., 2021), among others. The GradNorm (Huang et al., 2021) score considers the gradient information providing a direct measure of how uncertain the model is with respect to its parameters. In this work, we employ the GradNorm score for input examples to identify OOD data and cluster patterns, with the aim of effectively discovering and learning new classes from the data distribution.

## 2.3 Deep Active Learning

Active learning studies the problem of minimizing annotation cost while training high performance models. Active learning methods typically follows a sequential and adaptive procedure, where the algorithms first trains a model based on the annotated examples so far followed by annotating more examples selected from a large number of unlabeled examples. The algorithm strategically chooses from the unlabeled examples for annotation, typically relying on uncertainty, diversity or expected model change types of metrics.

Uncertainty based strategies chooses examples that are determined to be the most uncertain for the model trained on the labeled set thus far. This includes many of the traditional uncertainty metrics such as margin, entropy and confidence scores (Lewis & Gale, 1994; Tong & Koller, 2001; Balcan et al., 2006; Settles, 2009; Kremer et al., 2014; Wang et al., 2022b). More advanced approaches for deep learning also includes Bayesian

uncertainty estimation and adversarial training (Gal et al., 2017; Ducoffe & Precioso, 2018; Beluch et al., 2018). Diversity based strategies aim to choose a set of unlabeled examples that are maximally different in an embedding space. This is usually achieved by clustering and covering techniques or greedy optimization of some global submodular diversity objective (Sener & Savarese, 2017; Geifman & El-Yaniv, 2017; Bıyık et al., 2019; Citovsky et al., 2021; Bhatt et al., 2024). In this paper, we explore a wide range of these diversity-based methods in conjunction with OOD detection to discover new classes. Lastly, existing approaches also take a combination of uncertainty and diversity metrics, often predicting the expected model change if an example is labeled (Ash et al., 2019; 2021; Wang et al., 2021; Elenter et al., 2022; Mohamadi et al., 2022).

Active learning has recently focused much attention on the closely related field of open set learning (Karamcheti et al., 2021; Ning et al., 2022; Zhang et al., 2022; Bai et al., 2024; Safaei et al., 2024; Yang et al., 2024; Mao et al., 2024). In open set learning, the primary objective is to classify all examples of unknown concepts into a single out-of-distribution class. Our study, however, addresses the more challenging open-world scenario. In this setting, we aim for the learner to not only identify unknown concepts but also to further learn and classify these examples into their appropriate concept categories.

The challenge of missing categories during initial training commonly arises when class sizes follows long tail distributions. In this paper, we study this exact realistic setting of open world learning with long tail data distribution. A large body of deep active learning literature has focused on the imbalance dataset settings (Aggarwal et al., 2020; Kothawade et al., 2021; Emam et al., 2021; Coleman et al., 2022; Jin et al., 2022; Cai, 2022; Nuggehalli et al., 2023; Zhang et al., 2024b; Soltani et al., 2024; Zhang & Nowak, 2024). However, none of these work have studied the open world setting, where a subset of the classes are initially unknown to the algorithm. Below, we survey the few task-specific active learning algorithms that are proposed for open-world scenarios.

### 2.4 Open-World Active Learning

Overall, there has been few attempts in studying active learning under an open world setting. Ma et al. (2024) also studies the active learning setting where some classes do not have labeled examples. However, their paper assumes the knowledge of the total number of classes beforehand, making their study closer to the traditional active learning than truly addressing the open world challenge, where the total number of classes is agnostic to the learner. As we will discuss in Section 6, not knowing the number of classes poses a challenging tradeoff between exploring new classes and learning existing classes well.

Other existing literature shares our setting but targets specific tasks. Chen et al. (2023) propose OpenCRB for open-world 3D object detection, proposing an uncertainty-based scoring method that leverages the spatial distribution and density of data points for 3D point clouds. In the more general settings, this work essentially reduces to simple uncertainty methods, which we will show to be less effective than our methods in open world settings. Zamzmi et al. (2022) utilizes a simple diversity-based method, k-medoids clustering, in annotating a diverse set of images for echocardiography view classification. As we will show, our method that combines OOD detection and diversity-based methods is superior than only considering diversity alone. In Section 5.2.1 and Table 2, we provide details of the OOD scores we conduct experiments on.

## 3 Problem Setup

We consider a pool-based batched active learning setting in an open-world scenario. The learner has access to a large pool of unlabeled data $X = \{x_1, x_2, ..., x_N\}$, and the true label distribution is defined by an unknown ground truth function $f^* : X \to \{1, 2, ..., K\}$, where $K$ is the total number of classes in $X$. At the beginning the labeling process the labeled set $L$ is set to be empty. At each iteration $t$, the learner selects a small batch of $B$ unlabeled examples, $\{x_i^{(t)}\}_{i=1}^B \subseteq X$, from the pool. The learner observes the labels $\{f^*(x_i^{(t)})\}_{i=1}^B$ and adds these examples to $L$, the set of labeled examples available for training so far. In an open world setting, examples in $L$ may only cover a fraction of the classes, which we denote by $\mathcal{K}^t \subseteq [K]$, the known classes. After each batch query, the learner trains a $|\mathcal{K}^t|$-class classification model $f^t$ on $L$, which is used to inform the selection of the next batch of examples for labeling. Furthermore, we let $\widehat{p}^t(x)$ denote the softmax distribution score based on the classification model $f^t$.

This iterative process continues, with the goal of selecting the most informative examples to annotate in each round, minimizing the number of labeled examples needed to achieve good generalization performance. The generalization performance is measured by the balanced accuracy on all $K$ classes. If some of the classes do not have any annotated example, the accuracy for such classes are considered 0. Therefore, it is crucial to annotate a wide array of classes, while also learning the known classes well based on the annotated examples.

### 3.1 Out-of-Distribution (OOD) Scores

During each step $t$ of the active learning process, recall $\mathcal{K}^t$ represents the set of classes that have at least one labeled example. When training a model using the labeled dataset, examples from classes not in $\mathcal{K}^t$ (i.e., $[K]\backslash\mathcal{K}^t$) are considered out-of-distribution (OOD) by definition. Traditional OOD detection research focuses on identifying OOD examples within a test set. In this study, we use an OOD scoring function, denoted as $\Omega(x, f)$, where $x$ is an example and $f$ is any neural network classification model. This function $\Omega(x, f)$ provides a measure of how likely an example is to be in-distribution (ID) versus OOD. A higher OOD score suggests a greater likelihood that the example is OOD, which in turn is more likely to belong to an unknown class. We apply these OOD scoring techniques to determine which unlabeled examples are most likely to belong to unknown OOD classes.

## 4 Methodology

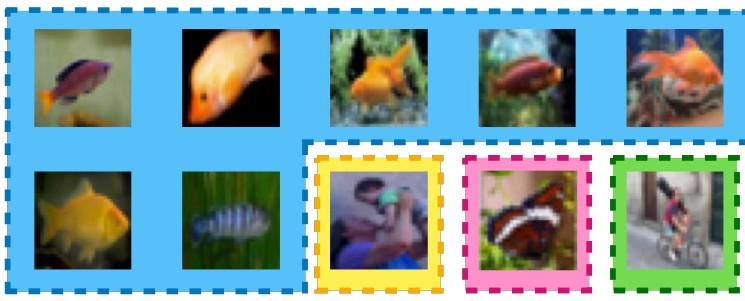

Figure 1: Visualization of highest OOD score unlabeled examples after training on three known classes in CIFAR-100. As we can see, relying only on OOD scores for selection will encourage annotation of unlabeled examples in a few (unknown) classes.

To address the open world active annotation problem, we focus on two key objectives: *i*) maximizing the selected examples' coverage of OOD classes; *ii*) optimizing the labeling budget to efficiently collect examples from these OOD classes. A straightforward approach would be to simply annotate examples with the highest OOD scores during each round of annotation. However, our analysis in Figure 1 reveals that examples with high OOD scores often cluster within a small subset of OOD classes, making this approach suboptimal for achieving broad coverage. This observation suggests the need for a diversity-based strategy to ensure our annotated examples span a wide range of concepts.

In Section 4.1, we present ALOE, a two-stage approach that combines OOD scoring with diversity-based active learning strategies. ALOE first employs clustering-based diversity methods to identify distinct groups of examples, then filters these clusters to retain only high-scoring OOD candidates for annotation. Our experiments show that applying diversity clustering before OOD filtering yields better results than the alternative strategy (see Section 4.2) that executes these steps in the reverse order.

### 4.1 ALOE: Our Actively Learning Algorithm Under Open-world Environments

In this section, we describe our proposed algorithm for querying unlabeled data in a pool-based batched active learning framework under open-world conditions. As detailed in Algorithm 1, our algorithm ALOE leverages OOD scores and k-means clustering method to ensure a balance between identifying novel OOD examples and maintaining diversity within the queried batch.

The algorithm proceeds iteratively for $T$ iterations, with each iteration $t$ comprising several key steps, as illustrated in Figure 2. Initially, a deep neural network undergoes is trained on the current labeled set $L_{t-1}$, resulting in an updated model $f^{t-1}$ that incorporates information from the newly labeled data. Subsequently, to ensure diversity in the selection process, the algorithm embeds the pool of unlabeled set $X'$ using the neural

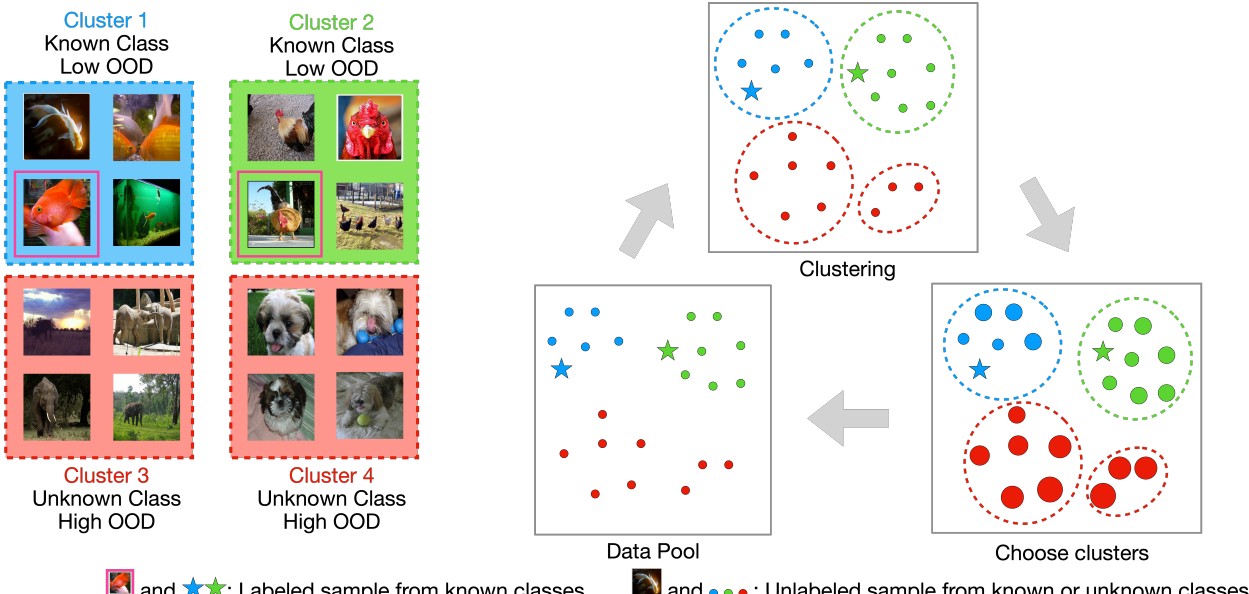

Figure 2: Illustration of our algorithm ALOE in Algorithm 1. Starting with a few labeled examples, the algorithm clusters all examples, ideally by their underlying classes. Each example's OOD score is calculated. In the bottom right plot, larger dots mean larger OOD score. Clusters with higher OOD ratios are prioritized for sampling to identify new classes. Labeled examples are then added to the training set, and the process iterates, expanding the labeled pool with each batch.

network features learned from $f^{t-1}$. The clustering method, k-means, is then applied to these embedded features. The number of clusters $2 \cdot \max(B, |\mathcal{K}_t|)$ is set so that we obtain a surplus of clusters to effectively filter out the in-distribution examples, where $B$ is the batch size and $|\mathcal{K}_t|$ is the number of annotated classes in step $t$. Empirically throughout our experiments, we find the multiplier value 2 to be a good and robust value. It can be imagined that other clustering methods can be used. While it will be discussed more in our experiments in Section 5.2.5, we stick to k-means when describing ALOE.

Following the clustering phase, we define the OOD cluster ratio as the proportion of predicted out-of-distribution (OOD) examples within each cluster, where predictions are made using an OOD scoring function $\Omega(x, f^{t-1}) : X \to \mathbb{R}$ defined in Section 5.2.1. To identify OOD examples, we establish a threshold at the 95%-TPR cutoff based on the in-distribution labeled examples. This threshold is commonly used in the OOD detection literature, and ensures at least 95% of the labeled examples are classified as ID. The clusters are then ranked based on their OOD cluster ratio. The top $B$ clusters are selected for further processing, and from each of these selected clusters, the example with the highest OOD score is chosen to form the final batch $X^{(t)}$ of examples for annotation.

The iteration concludes with an update phase where the labeled set $L_t$ is augmented with the newly labeled data, and these examples are correspondingly removed from the unlabeled pool $U_t$. Through this iterative process, the labeled set gradually expands to include a diverse range of both in-distribution (ID) and out-of-distribution (OOD) examples, thereby enhancing the classifier's ability to generalize to both known and novel classes that were unknown at the beginning of the annotation process.

## 4.2 Reverse ALOE: Alternative Query Strategy

In addition to the approach described above, we also considered an alternative method **reverse ALOE**. As the name suggests, the order of using OOD scores and clustering methods are flipped. OOD scores were used first to select an initial set of candidates in this algorithm. Similarly with ALOE, we calculated a threshold $\tau$, which ensures a 95% true positive rate (TPR) in labeled set $L_t$. The initial set are examples with OOD score larger than the threshold $\tau$.

---

**Algorithm 1** ALOE: ActiveLy Learning in Open-world Environments

---

**Input:** Pool of examples $X$, initial labeled set $L_1$ covering a subset of total classes $\mathcal{K}_1 \subseteq [K]$, initial unlabeled set $U_1 = X \backslash L_1$, number of iterations $T$, batch size $B$.

**Define:** OOD scoring function $\Omega : X \to \mathbb{R}$, which maps examples to their likelihood of being OOD based on the current labeled set $L$.

**for** $t = 2, \ldots, T$ **do**

    Fine-tune the large pretrained neural network on $L_t$ to obtain the model $f^{t-1}$.

    **Step 1: Embedding and Clustering for Diversity**

    Embed examples in unlabeled set $U_t$ using neural network features from $f^{t-1}$ and apply k-means clustering with $k = 2 \cdot \max(B, |\mathcal{K}_t|)$, resulting in clusters $X_c$ for each $c = 1, \ldots, k$.

    **Step 2: Identify Clusters with High OOD Probability**

    Rank clusters based on the ratio of likely OOD examples to cluster size. For each cluster $X_c$, the ratio of OOD examples is calculated as:

$$r_{\text{OOD}}^{(c)} = \frac{\sum_{x \in X_c} \mathbf{1}(\Omega(x, f^{t-1}) > \tau)}{|X_c|}$$

where $\mathbf{1}(\Omega(x, f^{t-1}) > \tau)$ indicates whether the OOD score $\Omega(x, f^{t-1})$ of an example $x$ exceeds the threshold $\tau$, which ensures a 95% true positive rate (TPR) in the labeled set $L_t$, and $|X_c|$ is the total number of examples in cluster $X_c$. Pick $B$ clusters with the highest OOD ratio.

    **Step 3: Query Examples from Chosen Clusters**

    From each of the $B$ selected clusters, pick the example with the highest OOD score. This forms the query set $X^{(t+1)}$.

    Annotate examples in $X^{(t+1)}$ and update the labeled and unlabeled sets:

$$L_{t+1} \leftarrow L_t \cup X^{(t+1)}, \quad U_{t+1} \leftarrow U_t \backslash X^{(t+1)}.$$

**end for**

**Return:** Fine-tune the pretrained model on the final labeled set $L_T$ to obtain the final classifier $f^{(T)}$.

---

These candidates were then clustered with methods described in Section 5.2.5 to ensure diversity among the queried examples with the cluster number equal to batch size. In each cluster, the sample with highest OOD score is queried to be labeled. However, this method performed poorly in our experiments (Section 5.2.2), as it did not select a super diverse set of examples, reducing the overall efficiency of the active learning process.

# 5 Experiment

## 5.1 Experiment Setup

**Dataset.** Open world challenges often arise from dataset imbalance with a large number of classes, where smaller classes are often unknown at the beginning of the annotation process. In this paper, we focus our experiment evaluations on common long tail imbalanced datasets with a large number of classes. Specifically, we utilize three image classification benchmark datasets, **CIFAR100-LT** (Alex, 2009), **ImageNet-LT** (Deng et al., 2009) and **Places365-LT** (Zhou et al., 2017). The distribution of the three long-tailed datasets are given in Appendix 7.

**Model.** For evaluation, we follow the latest LabelBench framework (Zhang et al., 2024a), while introducing the new open world setting with dynamic number of classes at each iteration. Specifically, we fine-tune the pretrained CLIP ViT-B32 image encoder (Radford et al., 2021) with a linear classification head attached. For every iteration of the active learning algorithm, the model is reinitialized to the pretraining checkpoint and finetuned end-to-end on all labeled examples thus far.

In our experiment, we utilize the cold starting approach by reinitializing the model from the CLIP model checkpoint after every iteration of annotation. Specifically, for labeled set $L_t$ with $|\mathcal{K}^t|$ known classes, we

attach a linear head on the CLIP image encoder model with $|\mathcal{K}^t|$ outputs. We then finetune this model on the labeled set $L_t$ for each iteration $t$.

**Metric.** To evaluate the performance of our method in the open-world active learning setting, we use two primary metrics: the number of annotated classes and the balanced accuracy. The number of annotated classes (denoted as $|\mathcal{K}_t|$ in Section 3) is crucial as missing certain categories could hinder the model's performance when deployed in practice. On the other hand, balanced accuracy gives a more wholistic view of the model's generalization performance. Specifically, given a test set of examples $(x'_1, y'_1), ..., (x'_M, y'_M)$ and a model $f$ mapping images to classes, the balanced accuracy is also known as the average recall

$$\text{ACC}_{\text{bal}} = \frac{1}{K} \sum_{k=1}^{K} \left[ \frac{1}{N_k} \sum_{i:y'_i=k} \mathbf{1}\{f(x'_i) = k\} \right],$$

where $K$ is the total number of classes, $N_k$ is the number of examples in class $k$. The balanced accuracy simply averages the classifier's recall of predicting each class. Note that if the classifier $f$ does not predict a certain class, the accuracy for that class is 0.

**Software and hardware.** Our method is implemented with PyTorch 2.2.0. All experiments are conducted on NVIDIA TITAN RTX for CIFAR100-LT and Places365-LT, and NVIDIA A100 for ImageNet-LT.

**Setup Summary.** The experimental setup for each dataset is summarized in Table 1. We conduct experiments over $T$ iterations, with a batch size of $B$ and an initial labeled set $L_1$ covering a subset of classes $\mathcal{K}_1$. For each of the following settings, we start with a small set of initially labeled classes ($|\mathcal{K}_1|$) to simulate a real-world open-world learning environment, where the model has limited knowledge of the complete label space at the beginning. The classes are chosen to be the largest $|\mathcal{K}_1|$ classes in size and the initial batch of labeled examples are distributed evenly across these classes.

Table 1: Experimental settings for each dataset

| Dataset | Initial #Classes ($|\mathcal{K}_1|$) | Total #Classes | #Iteration ($T$) | Batch Size ($B$) |
|---|---|---|---|---|
| CIFAR100 | 3 | 100 | 15 | 50 |
| ImageNet-1K | 10 | 1000 | 10 | 1000 |
| Places365 | 3 | 365 | 10 | 200 |
| CIFAR100 | 10 | 100 | 15 | 50 |
| CIFAR100 | 30 | 100 | 15 | 50 |
| CIFAR100 | 50 | 100 | 15 | 50 |

Additionally, in the last three rows of Table 1, we show the settings of ablation studies on CIFAR100-LT by varying the initial number of labeled classes ($|\mathcal{K}_1| = 10, 30, 50$) to assess the robustness of our algorithm across different scenarios. These studies highlight our method's ability to adapt to varying levels of initial knowledge while maintaining strong performance across different settings.

## 5.2 Main Results and Analysis

### 5.2.1 OOD Scoring Functions and Baseline Active Learning Algorithms

Before we present our experimental results, we introduce the OOD scoring functions and baseline active learning algorithms we use.

**OOD Scoring Functions:** In Table 2 we summarize the OOD scoring methods from previous literature, which are used throughout our research. **Energy** (Liu et al., 2020) score provides a global view of the uncertainty by aggregating information over all classes. These scores are particularly useful when the model outputs soft probabilities that span multiple classes. **Margin** (Scheffer et al., 2001) scores offer a more focused view by comparing only the top two predicted probabilities, making them sensitive to near-boundary decisions. **GradNorm** (Huang et al., 2021) considers the gradient information, which can provide a more direct measure of how uncertain the model is with respect to its parameters, especially in deep learning

models. **Mahalanobis distance** (Lee et al., 2018) is more geometric, considering how far a given example is from the expected distribution of a particular class. This distance metric is often used in embedding spaces. **Gradient-based** (Bai et al., 2024) score measures the model's sensitivity to input perturbations by calculating gradients, capturing how the model would change with slight variations.

Table 2: OOD scores used in the OOD score ablation study.

| OOD Scores | Definition | Description |
|---|---|---|
| **Energy Score** | $\Omega_{\text{Energy}}(x) = -\log \sum_{k=1}^{K} \exp\left(\widehat{p}_k^t(x)\right)$ | Aggregates log probabilities over all classes, providing a measure of overall uncertainty. |
| **Margin Score** | $\Omega_{\text{Margin}}(x) = \widehat{p}_{\max}^t(x) - \widehat{p}_{\text{second}}^t(x)$ | The difference between the highest and second-highest predicted probabilities, $\widehat{p}_{\max}^t(x)$ and $\widehat{p}_{\text{second}}^t(x)$, focusing on boundary cases. |
| **GradNorm** | $\Omega_{\text{GradNorm}}(x) = \|\nabla_\theta \mathcal{L}(x; f)\|_2$ | Norm of the gradient of the loss function $\mathcal{L}$ with respect to model parameters $\theta$, providing a measure of parameter-space uncertainty. |
| **Mahalanobis Distance** | $\Omega_{\text{Mahalanobis}}(z) = (z - \mu_c)^T \Sigma^{-1} (z - \mu_c)$ | Distance from the class mean of embeddings $\mu_c$ for corresponding class $c$. $\Sigma$ is the class covariance matrix. |
| **Gradient-Based Score** | $\Omega_{\text{Gradient}}(x) = \langle \nabla\mathcal{L}(x; f) - \overline{\nabla}, \mathbf{v} \rangle$ | Projection of the difference between sample gradients and their average onto the top singular vector $\mathbf{v}$ of gradient matrix, made by stacking gradients of all data. $\overline{\nabla}$ denotes the average gradient across dataset. |

**Baseline Active Learning Algorithms:** We evaluated our algorithm against several standard active learning baselines: **Random**, **Margin**, **Coreset**, **Galaxy**, **Badge**, **ProbCover**, **TypiClust**, and **EOAL**. **Random** is a straightforward baseline where examples are selected uniformly at random from the pool of unlabeled data, offering a comparison with non-strategic sampling. **Margin** (Scheffer et al., 2001) is an uncertainty-based sampling method that selects examples based on the difference between the top two predicted class probabilities. This approach aims to identify samples near decision boundaries, where the model is most uncertain. **Coreset** (Sener & Savarese, 2017) is a diversity-based approach that seeks a representative subset of the unlabeled pool by minimizing the maximum Euclidean distance between any point in the dataset and the nearest selected point in the subset. **Badge** (Ash et al., 2019) combines uncertainty and diversity by selecting examples based on gradient embeddings to cover informative examples. **Galaxy** (Zhang et al., 2022) is specifically designed for imbalanced scenarios, using a balanced sampling strategy to improve performance on classes with fewer labeled examples, making it well-suited for imbalanced datasets. **Prob-Cover** (Yehuda et al., 2022) adopts a covering-based approach to maximize probabilistic coverage over the data distribution, effectively identifying critical examples under a constrained labeling budget. It is important to note that ProbCover is highly sensitive to its hyperparameter. As active learning can only run onnce when deployed in practice, we selected a single value that is reasonably suitable across various scenarios. **TypiClust** (Hacohen et al., 2022) is a low-budget active learning method that prioritizes selecting samples from dense regions of the feature space, ensuring representative coverage even with limited labeling resources. Entropic Open-set Active Learning (**EOAL**) (Safaei et al., 2024) addresses open-set scenarios by employing entropy-based selection to handle both known and unknown classes effectively, ensuring robust performance in open-world settings. The time cost analysis of all baselines and ALOE is included in Appendix C.

### 5.2.2 Main Results on Multiple Datasets

We now present the results of our algorithm applied to CIFAR100-LT, ImageNet-LT, and Places365-LT. As mentioned before, the long-tailed distribution of classes is a well-known challenge in practical applications. Due to this issue, we often encounter open world learning scenarios where rare classes are likely to be missed during the initial annotation phase. In the following experiments, we receive annotations from only a small number of classes in the initial batch of annotation (three for CIFAR100-LT and Places365-LT, and ten for ImageNet-LT). In addition, our algorithm ALOE uses GradNorm for OOD scoreing function $\Omega$ and k-means for clustering method. We also include ablation studies around different initial number of classes, OOD scores, and clustering methods in the later sections. In Table 3 and Table 4, balanced test accuracy on the three datasets are shown. Number of annotated classe and the plot of the results are summarized in Table 5, Table 6 and Figure 8 in Appendix B.

Table 3: Balanced test accuracy on CIFAR100-LT and Places365-LT with different budget. (The standard error was calculated based on four trails.)

| Dataset | CIFAR100-LT | | Places365-LT | |
|---|---|---|---|---|
| Budget | 250 | 750 | 600 | 2000 |
| Random | $0.392_{\pm 0.006}$ | $0.553_{\pm 0.004}$ | $0.208_{\pm 0.007}$ | $0.275_{\pm 0.004}$ |
| Margin | $0.445_{\pm 0.006}$ | $0.643_{\pm 0.011}$ | $0.217_{\pm 0.005}$ | $0.287_{\pm 0.004}$ |
| Badge | $0.447_{\pm 0.006}$ | $\mathbf{0.652_{\pm 0.008}}$ | $0.231_{\pm 0.004}$ | $0.295_{\pm 0.004}$ |
| Galaxy | $0.436_{\pm 0.007}$ | $0.645_{\pm 0.008}$ | $0.231_{\pm 0.004}$ | $\mathbf{0.302_{\pm 0.005}}$ |
| Coreset | $0.449_{\pm 0.012}$ | $0.617_{\pm 0.010}$ | $0.238_{\pm 0.005}$ | $\mathbf{0.304_{\pm 0.003}}$ |
| ProbCover | $0.346_{\pm 0.010}$ | $0.442_{\pm 0.004}$ | $0.152_{\pm 0.007}$ | $0.247_{\pm 0.005}$ |
| TypiClust | $0.376_{\pm 0.015}$ | $0.509_{\pm 0.004}$ | $0.199_{\pm 0.005}$ | $0.275_{\pm 0.004}$ |
| EOAL | $0.368_{\pm 0.005}$ | $0.578_{\pm 0.005}$ | $0.211_{\pm 0.007}$ | $0.277_{\pm 0.005}$ |
| ALOE(Ours) | $\mathbf{0.479_{\pm 0.007}}$ | $\mathbf{0.650_{\pm 0.04}}$ | $\mathbf{0.254_{\pm 0.004}}$ | $\mathbf{0.310_{\pm 0.008}}$ |

Table 4: Balanced test accuracy on ImageNet-LT with different budget. (The standard error was calculated based on four trails.)

| Dataset | ImageNet-LT | |
|---|---|---|
| Budget | 3000 | 10000 |
| Random | $0.391_{\pm 0.002}$ | $0.541_{\pm 0.003}$ |
| Margin | $0.424_{\pm 0.006}$ | $0.562_{\pm 0.002}$ |
| Badge | $0.434_{\pm 0.004}$ | $0.581_{\pm 0.002}$ |
| Galaxy | $0.438_{\pm 0.006}$ | $\mathbf{0.631_{\pm 0.001}}$ |
| Coreset | $0.351_{\pm 0.005}$ | $0.525_{\pm 0.002}$ |
| ALOE(Ours) | $\mathbf{0.517_{\pm 0.004}}$ | $\mathbf{0.617_{\pm 0.003}}$ |

As shown in Table 3 and Table 4, our method ALOE significantly outperforms all baseline methods in both balanced accuracy and the number of newly discovered classes. Starting with only three initially labeled classes, our method quickly expands the set of annotated classes, maintaining a higher discovery rate throughout the iterations. By leveraging the combination of OOD detection and diversity-based sampling, ALOE is able to efficiently explore unknown classes while ensuring strong performance on the known classes. On ImageNet-LT, ALOE saves 70% of annotation cost to achieve the same accuracy comparing to random sampling. We also note that while some baselines occasionally match ALOE's performance, they struggle with consistency across different datasets. In contrast, ALOE consistently performs best among all methods on all datasets.

We also experimented with Reverse ALOE, which doesn't show sufficient label efficiency comparing to other baselines. The Reverse ALOE result shown in Figure 8 uses the combination of GradNorm OOD score and Coreset clustering method, which behaves best in all combinations. While still competitive with some baselines, Reverse ALOE does not match the performance of ALOE. This performance gap likely stems from conducting OOD filtering first, which eliminates many classes of out-of-distribution examples early in the

process. As a result, Reverse ALOE works with a less diverse set of examples compared to ALOE, which preserves diversity by clustering examples before performing OOD filtering.

Interestingly, the advantage of ALOE becomes more pronounced with datasets that have a larger number of classes, espiecially on earlier iterations of experiment. ImageNet-LT, which has the highest number of classes (1,000), demonstrates a noticeable gap between ALOE and other baselines. For CIFAR100-LT and Places365-LT, with 100 and 365 classes respectively, ALOE still performs well, although the difference is less striking. This trend reflects the inherent nature of ALOE, which excels at identifying new classes.

Another interesting observation is, on ImageNet-LT, ALOE outperforms all baselines until the annotation budget around 7000. However, when most classes have been discovered, GALAXY slightly outperforms ALOE. Similarly, on Places365-LT, ALOE achieves strong performance in terms of balanced test accuracy, but did not find new classes as fast as Badge in later iterations. This reveals an interesting paradigm shift in active learning between discovering new categories/clusters of examples and learning the existing categories/clusters of examples. We discuss this phenomenon further in Section 6 as a novel challenge through the lens of open world scenarios.

### 5.2.3 Ablation Study of Initial Number of Annotated Classes

In Figure 3, we analyze the robustness of our algorithm by varying the initial number of annotated classes on CIFAR100-LT. We experiment with three settings, including the initial number of annotated class being 10, 30 and 50 respectively. As observed in this experiment, ALOE consistently performs the best among all algorithms and is robust to different initial number of classes. It is also worth noting that despite maintaining strong performance, the improvement over the second best algorithm narrows slightly compare to the experiment in the previous section, where only three classes are initially labeled. This observation also corroborates our discussion on the paradigm shift between discovering more classes and learning existing classes. As mentioned before, we further discuss this challenge in Section 6.

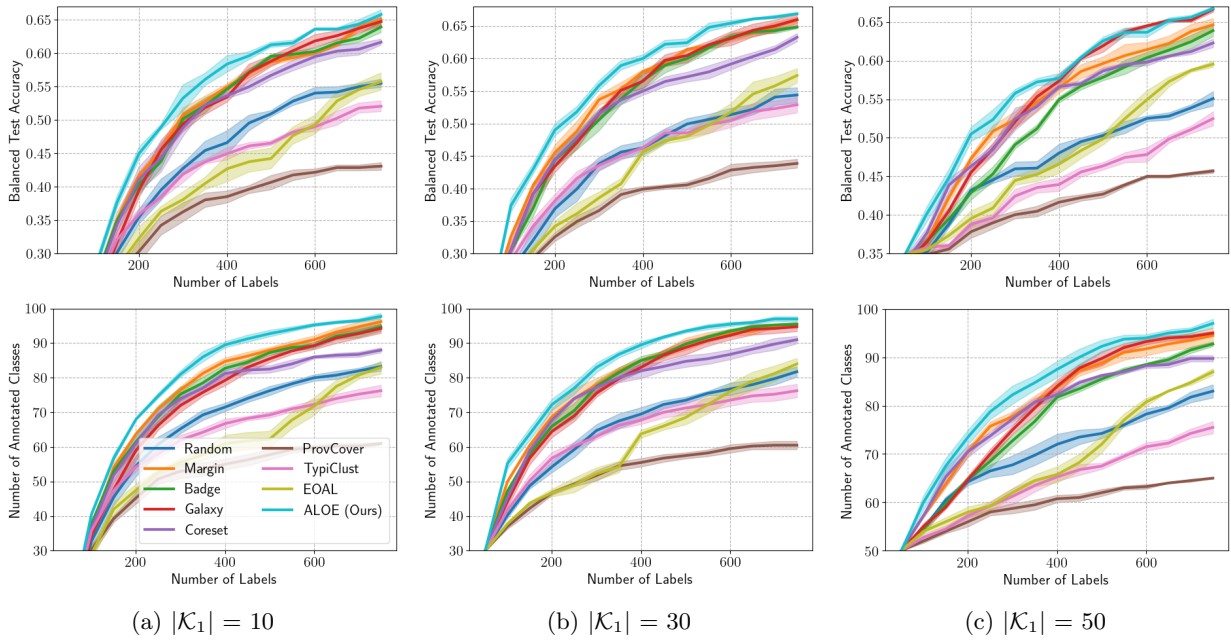

(a) $|\mathcal{K}_1| = 10$    (b) $|\mathcal{K}_1| = 30$    (c) $|\mathcal{K}_1| = 50$

Figure 3: With different initial number of annotated classes, balanced test accuracy and number of annotated classes of CIFAR100-LT. (All subfigures share the same legend of the left bottom one. Shaded region represent the standard error conducted across four trials.)

### 5.2.4    Ablation Study of OOD Scores

Figure 4 shows that different OOD scores affect the performance of our algorithm. On CIFAR100-LT, Energy and GradNorm scores perform similarly. However, we used GradNorm to carry out our main experiments in Section 5.2.2, since it is a direct reflection on the uncertainty of the model, espiecially deep learning models.

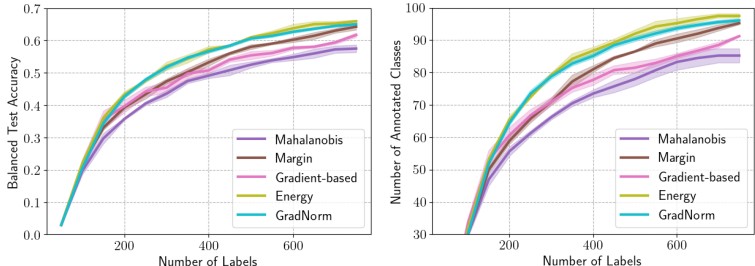

Figure 4: Ablation experiment for using different OOD scores $\Omega(x, f)$ in ALOE on CIFAR100-LT. (Shaded region represents the standard error conducted across four trials.)

### 5.2.5    Ablation Study of Clustering Methods

We also investigated various clustering methods. We experimented with a popular clustering method used in prior active learning algorithms. For instance, TypiClust (Hacohen et al., 2022) employs **k-means**, Coreset (Sener & Savarese, 2017) utilizes **k-center**, and BADGE (Ash et al., 2019) uses **k-means++**. We also implemented **gaussian mixture models** and **mini-batch K-means** as the clustering subprocedure. Both of these are more computationally efficient than simple k-means, and obtain similar performance as the procedure using k-means. As shown in Figure 5, this indicates that our algorithm is not highly sensitive to the choice of clustering algorithm. In our implementation, we used k-means despite its relatively higher computational cost compared to other clustering methods. However, in most active learning scenarios, the selection cost is significantly lower than the cost of training the neural network. In the rare cases where k-means becomes impractical, we recommend practitioners consider alternatives such as Gaussian Mixture Models (GMM) or mini-batch k-means.

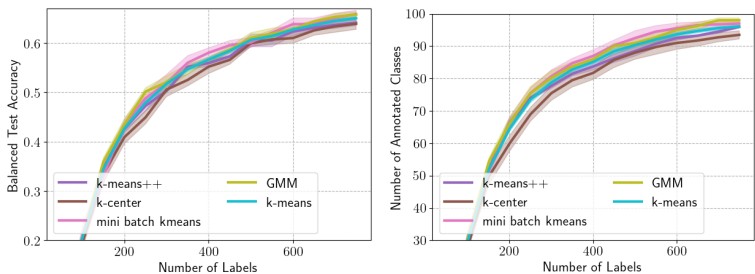

Figure 5: Ablation experiment for clustering methods on CIFAR100-LT with GradNorm OOD score. (Shaded region represent the standard error conducted across four trials.)

## 6    Balancing Novel Class Discovery and Known Class Learning

When running ALOE further on a larger amount of annotation budget, our experiments (Figure 6) demonstrate an intriguing phenomenon – baseline methods can even achieve slightly better performance than ALOE once most classes have been identified. Similarly, as mentioned in Section 5.2.3, on CIFAR100-LT, when the number of initial classes increases, the improvement gap of using ALOE narrows. These findings are expected, as ALOE uniquely prioritizes annotating diverse OOD classes, while baseline approaches focus on learning known ID classes. While this suggests the potential for an algorithm that excels at both tasks, achieving this dual objective presents significant challenges.

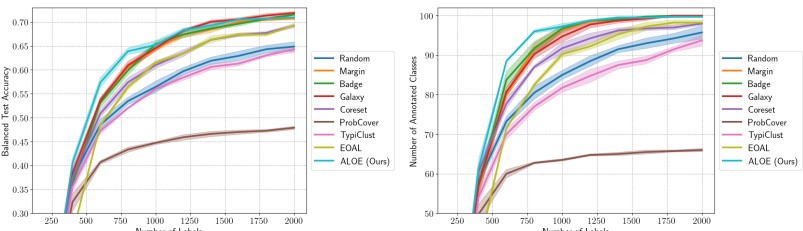

Figure 6: Balanced test accuracy and number of annotated classes on CIFAR100-LT when batch size $B = 200$ and number of iteration $T = 10$. (Shaded region represent the standard error conducted across four trials.)

This challenge arises because the total number of classes is unknown to the learner. Even after most classes have been discovered in our ImageNet-LT and Places365-LT experiments, the learner cannot determine whether significant numbers of classes remain undiscovered. Furthermore, our experiments demonstrate that ALOE, while perform the best during the class discovery phase, may not be the ideal strategy for learning in-distribution (ID) classes. This reveals a fundamental tradeoff in open-world active learning: the need to balance annotation between exploring novel classes and consolidating knowledge of existing ones.

Future research could approach this challenge from a theoretical perspective by developing strategies that dynamically alternate between exploration and consolidation. One promising direction is an iterative approach that first ensures discovery of classes above a certain size threshold, followed by focused learning of these identified classes. This process could then repeat with progressively lower the size thresholds, ultimately yielding a model that performs well across classes of varying frequencies.

## 7 Conclusion

In this paper, we propose ALOE, a novel active learning algorithm specifically designed for open-world scenarios. Our approach leverages a two-stage process that combines diversity-based sampling with out-of-distribution detection, enabling the discovery and learning of new classes in a dynamic environment. Through experiments on three long-tailed datasets, ALOE demonstrates a consistent and clear advantage over existing baselines in both balanced accuracy and class discovery. The results also highlight a crucial tradeoff: while identifying new classes is important, effectively learning from previously discovered classes presents a competing challenge. This remains an unresolved issue that warrants significant research attention.

**Acknowledgement**

This work has been supported by ONR MURI grant N00014-20-1-2787.

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

## A Long Tail Datasets

**CIFAR100.** CIFAR100 contains 60,000 images across 100 classes, with 500 training images and 100 testing images per class. To create a long-tailed version of CIFAR100, we use an exponential distribution where the number of examples per class $N_i$ is given by:

$$N_i = N_0 \alpha^{\frac{i}{n}},$$

where $n$ is the total number of classes, $N_0$ is the number of examples in the most frequent class, and $\alpha$ is the imbalance factor. In our experiments, we set $\alpha = 0.01$, creating a highly imbalanced version of the dataset.

**ImageNet.** We use the ImageNet-1k subset, containing 1.28 million training images across 1,000 classes. Similar to CIFAR100, we create a long-tailed version of ImageNet using an exponential distribution with the same formula. Again, we set $\alpha = 0.01$ to generate the imbalance, resulting in a diverse distribution of images per class.

**Places365.** Places365 is a large-scale scene recognition dataset consisting of over 10 million images across 365 classes, designed for training and evaluating models on a wide variety of scene categories, including indoor and outdoor environments. Each class has up to 5,000 images for training, providing a balanced dataset for scene classification tasks.

The Places365-LT we used in our experiments are based on the distribution described in Liu et al. (2019). This version follows an exponential distribution, where the number of examples per class ranges from a maximum of 4,980 images to as few as 5 images. The total is 62500 images. The distribution was implemented by referring to the publicly available code associated with the paper.

The distribution of the three long-tailed datasets is visualized in Appendix Figure 7.

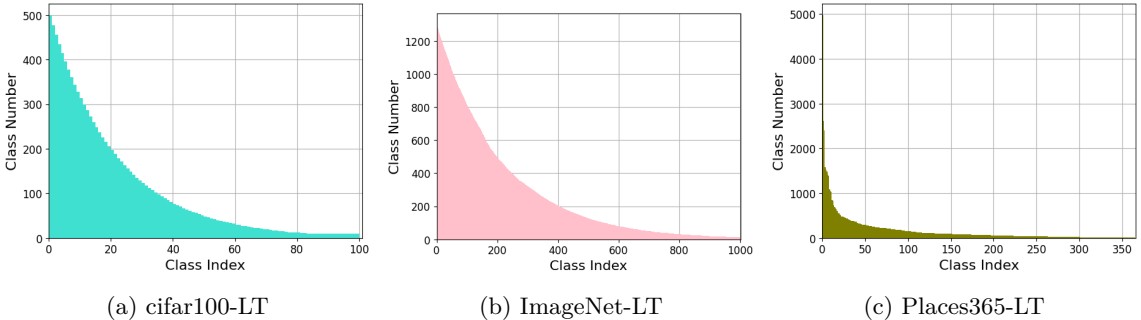

(a) cifar100-LT       (b) ImageNet-LT       (c) Places365-LT

Figure 7: Number of images in each class of train datasets

## B Experiments Results

## C Per-batch Running Time Analysis of ALOE and Baselines

We analyze the per-batch running time of ALOE in comparison to the mentioned baselines. Recall that $B$ is the batch size, $N$ is the pool size, $K$ is the number of classes, $Q$ is the forward inference time of a neural network on a single example, and $d$ is the feature dimension. Random Sampling is computationally the simplest, with a time complexity of $O(B)$, as examples are chosen uniformly at random without any additional computation. Confidence Sampling methods, such as Margin Sampling and EOAL, require $O(QN + KN + B \log N)$, where $O(QN)$ comes from forward passes on the pool, $O(KN)$ from computing uncertainty scores (e.g., margins or entropy), and $O(B \log N)$ from selecting the top $B$ uncertain examples.

Diversity-focused methods like Badge and Coreset involve more computational overhead. Badge combines uncertainty and diversity through gradient embeddings, requiring $O(QN + Nd^2 + Bd^3)$, where $O(Nd^2)$ arises

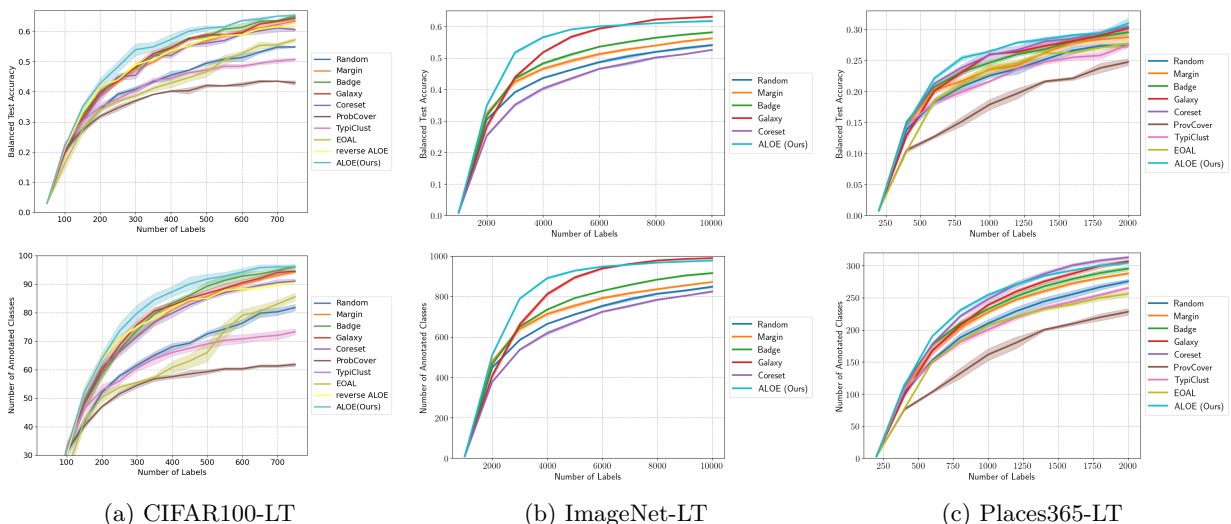

Figure 8: Balanced test accuracy and number of annotated classes on CIFAR100-LT, ImageNet-LT and Places365-LT with all baselines and our algorithms. Reverse ALOE result shown on CIFAR100-LT. (Shaded region represent the standard error conducted across four trials.)

Table 5: Number of annotated classes on CIFAR100-LT and Places365-LT with different budget. (The standard error was calculated based on four trails.)

| Dataset | CIFAR100-LT | | Places365-LT | |
|---|---|---|---|---|
| Budget | 250 | 750 | 600 | 2000 |
| Random | $57.75_{\pm0.63}$ | $81.75_{\pm1.11}$ | $188.00_{\pm6.36}$ | $275.50_{\pm3.30}$ |
| Margin | $66.75_{\pm1.31}$ | $94.75_{\pm1.31}$ | $203.75_{\pm4.61}$ | $287.75_{\pm2.02}$ |
| Badge | $66.75_{\pm0.75}$ | $\mathbf{96.00_{\pm0.82}}$ | $209.75_{\pm2.87}$ | $295.25_{\pm3.38}$ |
| Galaxy | $68.50_{\pm2.50}$ | $94.50_{\pm0.29}$ | $207.25_{\pm7.08}$ | $\mathbf{306.50_{\pm2.33}}$ |
| Coreset | $66.25_{\pm1.70}$ | $91.0_{\pm0.71}$ | $\mathbf{221.25_{\pm2.53}}$ | $\mathbf{312.75_{\pm1.93}}$ |
| ProbCover | $51.50_{\pm0.96}$ | $61.75_{\pm0.63}$ | $132.25_{\pm8.15}$ | $228.00_{\pm5.05}$ |
| TypiClust | $56.00_{\pm1.08}$ | $73.25_{\pm1.03}$ | $181.00_{\pm4.64}$ | $265.00_{\pm2.48}$ |
| EOAL | $53.75_{\pm0.95}$ | $85.5_{\pm1.32}$ | $182.25_{\pm1.80}$ | $256.00_{\pm3.92}$ |
| ALOE(Ours) | $\mathbf{73.5_{\pm0.94}}$ | $\mathbf{96.1_{\pm0.51}}$ | $\mathbf{230.50_{\pm1.80}}$ | $304.25_{\pm2.17}$ |

Table 6: Number of annotated classes on ImageNet-LT with different budget. (The standard error was calculated based on four trails.)

| Dataset | ImageNet-LT | |
|---|---|---|
| Budget | 3000 | 10000 |
| Random | $584.75_{\pm2.90}$ | $847.75_{\pm5.02}$ |
| Margin | $639.50_{\pm10.73}$ | $871.00_{\pm1.96}$ |
| Badge | $650.75_{\pm4.50}$ | $915.25_{\pm3.12}$ |
| Galaxy | $660.50_{\pm12.84}$ | $\mathbf{989.0_{\pm1.29}}$ |
| Coreset | $536.75_{\pm5.02}$ | $824.25_{\pm5.19}$ |
| ALOE(Ours) | $\mathbf{789.25_{\pm4.66}}$ | $\mathbf{977.00_{\pm2.04}}$ |

from gradient embedding computation and $O(Bd^3)$ from k-means++ clustering. Coreset solves a k-center problem, leading to a complexity of $O(QN+N^2d)$ in its exact form, although approximation methods reduce this to $O(QN + BNd)$. TypiClust is simpler, with $O(QN + BN)$, and ProbCover incurs $O(QN + N^2d)$ due to pairwise coverage probability computations.

For active learning algorithms in practice, time complexity is often not a critical bottleneck because $O(QN)$, representing the cost of forward passes on the pool, dominates all other terms. This ensures that methods like ALOE remain computationally efficient while incorporating both diversity and OOD detection to achieve superior performance in open-world active learning scenarios.

