# OpenReview forum: "Deep Active Learning in the Open World"
_TMLR — Accepted by TMLR_

### Review · Reviewer_58BP · 2024-11-26

**Summary Of Contributions:**

The Out-Of-Distribution (OOD) problem is interesting and highly relevant to real-world applications. The presentation is clear and easy to follow. However, I have some concerns:

Out-of-Distribution Details: The out-of-distribution (OOD) setup requires further clarification. For instance, the problem setup appears to be online, where data with different classes arrives in batches or one by one. How is the target set defined in a streaming scenario? For example, if a machine learning model assigns low probabilities to all classes for a sample, this could indicate that the sample belongs to a new class or that the model struggles to identify an existing class. Is there a standardized approach for mapping such samples to a new class?

Computational and Memory Complexity: What are the computational and memory complexity of the proposed algorithm? Specifically, does the use of k-means in each iteration limit its applicability to large-scale problems?

Practical Scenario Analysis: Is there any analysis of the practical scenarios where different OOD scores from Table 2 are applicable? A deeper examination of these cases would enhance understanding.

Experimental Results: While the main contribution of this work is the proposed algorithm for handling out-of-distribution, the experimental results do not convincingly demonstrate its advantage over existing methods. For example, as shown in Figure 3, the proposed methods, ALOE w/Energy and ALOE w/Gradnorm, often overlap with other methods. It is evident from the figures that the gap between the proposed methods and other approaches is not significant. If the authors wish to assert that the improvement is substantial, providing a table or conducting a relevant significance analysis is necessary.

**Audience:**

Yes

**Broader Impact Concerns:**

None.

**Claims And Evidence:**

No

**Requested Changes:**

Clarify the out-of-distribution (OOD) setup, including details such as how new labels are detected. Is there a standard definition being followed, or does this work consider alternative practical scenarios?

Include an analysis of the computational and memory complexity of the proposed method.

Provide insights into when and how to use the different OOD scores presented in Table 2.

Present experimental results in tables or include a detailed analysis to effectively validate the proposed method's effectiveness.

**Strengths And Weaknesses:**

The Out-Of-Distribution (OOD) problem is interesting and highly relevant to real-world applications.

While the primary contribution lies in its practical applicability, the complexity of the proposed method is not addressed. Furthermore, the experimental results, as shown in the figures, do not effectively demonstrate the advantages of the proposed approach.

---

> ### Author Response · Authors · 2024-12-14
>
> We would like to thank the reviewer for the insightful review.
>
> There appears to be a fundamental misunderstanding of our setup. To clarify, we are studying the pool-based active learning setting, where all unlabeled examples are accessible at the beginning. Active learning studies how to iteratively label more unlabeled examples. As more examples get labeled, more classes are discovered. When training on the latest labeled set of examples, we dynamically adjust the neural network architecture to accommodate for all of the discovered classes so far. We do not rely on OOD detection to create new classes. Instead, we use OOD detection to find unlabeled examples that are likely to be from new classes. The examples chosen by our algorithm are then annotated, which reveals the true label.
>
> In Table 3 of 5.2.3, we summarized the accuracy of the main experiments. In terms of more baselines, we added ProbCover, TypiClust, MaxHerding and EOAL. The four baselines are relatively new, with two of them from 2024. Despite their recency, we do not observe improvements in the novel setting of active open world learning. In fact, some of these algorithms perform worse than random. We believe this further suggests the challenges posed in our setting. Overall, ALOE, despite a simple algorithm, is much more effective than all these existing baselines. We will also include the full results on ImageNet for these algorithms in the camera-ready version.
>
> We have also added the computational complexity section as Appendix C.
>
> For better visualization of the performance gap, we have chosen to remove the ALOE w/ Energy baselines. As we have observed across all of our experiments, GradNorm is a superior OOD score. In addition, ALOE w/GradNorm outperforms other baseline methods by at least two times standard error, as summarized in our Tables.

---

> > ### Comment · Reviewer_58BP · 2024-12-17
> >
> > The manuscript should include additional clarification. Dynamically adjusting the neural network architecture, if necessary, is non-trivial. If the adjustment involves changing the parameters of the final mapping layer to accommodate a different number of classes, it should be explicitly stated how parameters in the preceding layers can be reused. Providing a clear and detailed explanation will help readers reproduce the work.
> >
> > Since this is a practical study, it would also be beneficial to include relevant natural language processing (NLP) tasks and cite related works, such as
> >
> > [1] J. Wang, J. Shen, X. Ma, Andrew Arnold. Uncertainty-based active learning for reading comprehension. TMLR, 2022.
> > [2] K. Siddharth, K Ranjay, FeiFei Li, M. Christopher, Mind Your Outliers! Investigating the Negative Impact of Outliers on Active Learning for Visual Question Answering, ACL, 2021.

---

> > > ### Author Response · Authors · 2024-12-22
> > >
> > > Thank you again for your feedback. We have addressed this comment in the model section of the experiment setup of our paper. Could you please let us know if this has resolved your concerns?

---

> > > > ### Comment · Reviewer_58BP · 2025-01-05
> > > >
> > > > I appreciate that the authors have addressed several concerns. However, there are some questions that the authors had promised to resolve in the discussion but appear to be unaddressed in the current version of the manuscript, such as including dynamically changing the model prediction class size in the manuscript, including the suggested related NLP works about active learning.

---

> > > > > ### Comment · Reviewer_PXtB · 2025-01-05
> > > > > **Clarification**
> > > > >
> > > > > Could the authors provide a clarification on the points raised by reviewer 58BP?

---

> > > > > > ### Author Response · Authors · 2025-01-05
> > > > > >
> > > > > > Thank you both for checking our manuscript and the continued support in help improving our paper.
> > > > > >
> > > > > > We did add the details regarding resizing the neural network. We have now highlighted the details in red on page 7. We have also added the two papers suggested by the reviewer and those are highlighted in red in the related work section.

---

> > > > > > > ### Comment · Reviewer_58BP · 2025-01-06
> > > > > > >
> > > > > > > I would like to thank the authors for addressing my concerns. I recommend that the authors review the reference formatting. For example, in the reference 'Uncertainty-based active learning for reading comprehension,' the journal name (TMLR) is missing. Once the references are correctly formatted, I have no further concerns and would recommend the publication of this work.

---

> > > > > > > > ### Author Response · Authors · 2025-01-06
> > > > > > > >
> > > > > > > > Thank you for pointing it out. We have fixed it.

---

> ### Author Response · Authors · 2024-12-17
>
> Thank you for the suggestion. We will include some of the following details regarding how to dynamically change the model prediction class size. Before that, we would first like to note that even in fixed class sizes, existing active learning approach mostly take a retraining approach after every new batch of annotation. That is, let $L_t$ denote the labeled set up till the $t$-th round, despite the current model has already been trained on $L_{t-1} \subset L_t$, the model for the next round is retrained from scratch or finetuned from the same pretraining checkpoint. This is referred to as the cold start approach. Continually training $L_t$ based on the model trained on $L_{t-1}$ is referred as the warm starting approach. It is commonly known in the active learning community that warm starting oddly performs way worse than cold starting. Few papers have provided an effective warm starting training strategy, and it is not clear if they will ever work for the open world setting. Therefore, in this paper, like in most other active learning papers, we take a cold starting problem, always starting from the pretrained CLIP model checkpoint. It is then very straightforward. For labeled set $L_t$ with $|\mathcal{K}^t|$ known classes, we attach a linear head on the CLIP image encoder model with $|\mathcal{K}^t|$ outputs. We then finetune this model on the labeled set $L_t$ for each iteration $t$.
>
> Thank you for your suggestion on the NLP papers. We have cited several NLP active learning papers in our paper and will also include these. Similar to prior deep active learning work, we have mostly adopted the same benchmarks, which have been mostly based on vision datasets. We think given the amount of evidence, it is clear that our algorithm performs well in the open world setting. In addition, most NLP classification tasks have less value and impact these days, as they can be effectively handled by LLMs. Many vision classification tasks still remain challenging today, so we primarily focus on vision tasks in this paper.

---

### Review · Reviewer_dXC5 · 2024-11-29

**Summary Of Contributions:**

The paper explores open-world machine learning, focusing on effectively handling out-of-distribution (OOD) data. Specifically, it introduces an active learning approach that effectively discovers an unknown number of OOD classes. The study compares this method to common deep active learning strategies, highlighting a paradigm shift in active learning: from discovering new classes of examples to refining the understanding of existing classes.

**Audience:**

Yes

**Broader Impact Concerns:**

None.

**Claims And Evidence:**

Yes

**Requested Changes:**

None for now, but please see points under weaknesses above.

Note: I found some small typos, such as missing space before citations in the introduction. Also, there seems to be some space missing directly under the description of Figure 2.

**Strengths And Weaknesses:**

Strengths:

* The paper is very clear and well written.
* The proposed method is simple and easy to implement (Alg. 1).
* The proposed method seems to perform well.

Weaknesses:

* W1: The experimental setup is solid overall, but I believe the paper would benefit significantly from more extensive experiments. Specifically, I expected additional results in the Appendix, where you investigate the paradigm shift discussed in Section 6 in much more detail. For instance, would similar observations hold if $f^t$ was not assumed to be pretrained? Expanding on this would provide a more comprehensive analysis (see point below also).
* W2: There is a large body of work addressing proper evaluation of deep active learning methods (see, e.g., [1, 2]). A key insight from these studies is that the performance of different query strategies can vary significantly depending on factors such as: 1) the number of random seeds, and 2) the initialization of the prediction model $f^t$ (e.g., whether it is pretrained or not). Considering that this paper includes only four random repetitions and assumes a pretrained model, this raises some concerns. That said, I would not view this as grounds to reject a paper on (deep) active learning, provided the proposed method is sound and demonstrates strong performance. Nonetheless, I am curious to hear the authors' perspective on this.
* W3: The paradigm shift discussed in Section 6 appears to primarily occur between the proposed method (ALOE) and the baseline Galaxy. I was hoping for a more detailed discussion to explain why this is the case, as it could provide valuable insights into the underlying reasons for this phenomenon. Why do we not, as far as I can tell, observe the paradigm shift with the other baselines?
* W4: I wonder if the reason your method occasionally performs worse in later iterations might be due to a higher tendency to select outliers—specifically, those that are uninformative due to noise. This could negatively impact performance, particularly in datasets with a significant number of outliers. Do you think this could be the case? I suspect the extent of this effect may depend on the value of $k$. If so, an experimental study that verifies this would be interesting.
* W5: In Figure 3c, why do you not continue until all classes are discovered for the Places365-LT dataset?
* W6: It may be good to be a bit more detailed regarding the implementation of the baselines. In particular, how are distances computed for coreset (i.e., in what feature space)?

**References**:

[1] Carsten T. Lüth, Till J. Bungert, Lukas Klein, Paul F. Jaeger:
Navigating the Pitfalls of Active Learning Evaluation: A Systematic Framework for Meaningful Performance Assessment. NeurIPS 2023.

[2] Thorben Werner, Johannes Burchert, Maximilian Stubbemann, Lars Schmidt-Thieme:
A Cross-Domain Benchmark for Active Learning. CoRR abs/2408.00426 (2024)

---

> ### Author Response · Authors · 2024-12-14
>
> We would like to thank the reviewer for the insightful feedback. We’ve made the following changes as you requested.
>
> W1: Regarding the issue of training models from scratch, we have previously conducted some experiments on ResNet, and the results were generally inconclusive. We believe this is primarily because, in our experiments, apart from the initial classes, the newly discovered classes often only contain one or two samples. For ResNet, this makes training nearly impossible.
>
> W2: We decided to do four runs based on previous research on active learning. Across our experiments, we can already obtain a solid standard error with four runs. Moreover, from our results, it is clear that ALOE outperforms other baselines by at least 2x standard error, showing a statistically significant advantage.
>
> W3: We believe the reason Galaxy is the best among all baselines, especially on ImageNet is that it is designed for imbalanced datasets. As our setting investigates the algorithms’ performances on long-tail imbalanced datasets, Galaxy expectedly performs better than other baselines, especially after most classes have been discovered. Other baselines didn’t focus on the imbalanced datasets.
>
> W4: While it is true that our algorithm is searching for outliers, it is a double sword. It is clearly helpful when there are still a lot of classes to be discovered. As the reviewer noted, it is not helpful once most classes have been discovered. However, from the learner’s perspective, it does not know which scenario this is under. We have previously ran a few experiments with $k$ being the budget, which hurts the performance early on, while improves the performance once most classes have been discovered. As we have noted in the last section, balancing the two effects is still largely an open problem.
>
> W5: In the places dataset, there are 365 classes in total. We chose our budget by stopping around when most classes have been discovered. This is because after most classes have been discovered, the problem turns into a regular active learning setting, and will not demonstrate the unique challenges posed by the open world setting.
>
> W6: We added this detail in the paper.

---

> > ### Comment · Reviewer_dXC5 · 2024-12-21
> >
> > Thank you for addressing my comments and those raised by the other reviewers. I am satisfied with your rebuttal, which adequately clarifies the points I raised. I also reviewed the responses to other reviews and found no additional concerns.

---

### Review · Reviewer_PXtB · 2024-11-30

**Summary Of Contributions:**

The authors propose an active learning algorithm designed for open-world settings. During the active learning process the proposed method embeds samples by the currently trained neural network. These samples are clustered using k-means. Subsequently, the top-N clusters are selected based on the OOD score of the samples within the clusters. Lastly, the samples with the highest OOD scores are queried.

The authors demonstrate the effectiveness of their algorithm on several (longtail) datasets and additionally conduct ablation studies concerning the design choices (such as the used OOD metric or the clustering algorithm).

**Audience:**

Yes

**Claims And Evidence:**

Yes

**Requested Changes:**

- Could the authors additionally provide a table with the final accuracies of the main result in the appendix?
- More recent baselines would strengthen the claims
- Could the authors provide a computational complexity comparison across methods?

**Strengths And Weaknesses:**

**Strengths**
- Statements are mostly backed up with experimental results
- Design choices of the algorithm are ablated

**Weaknesses**
- The motivation behind the algorithm is very high level. It could be difficult for the community to extract some takeaways from this paper, which can be built upon.
- Baselines are rather old
- Differences between the methods tend to be small, and the way the results are presented makes it difficult to compare them accurately.
- What is the computational overhead of the algorithm? K-Means does not scale well with a very high number of samples (realistic scenarios in open-world active learning). The evaluation of lightweight clustering alternatives to improve scalability would be an interesting ablation study

More recent methods to compare to (I do not have a specific preference. I am sure there are even more recent methods available):

Dongmin Park, et al., "Meta-query-net: Resolving purity-informativeness dilemma in open-set active learning", NeurIPS 2022

Pan Du, et al., "Contrastive active learning under class distribution mismatch", TPAMI 2023

Yang Yang, et al., "Not all out-of-distribution data are harmful to open-set active learning", NeurIPS 2023

---

> ### Author Response · Authors · 2024-12-14
>
> We would like to thank the reviewer for the insightful feedback. We’ve made the following changes as you requested.
>
> In Table 3 of 5.2.3, we summarized the accuracy of the main experiments. In terms of more baselines, we added ProbCover, TypiClust, MaxHerding and EOAL. The four baselines are relatively new, with two of them from 2024. Despite their recency, we do not observe improvements in the novel setting of active open world learning. In fact, some of these algorithms perform worse than random. We believe this further suggests the challenges posed in our setting. Overall, ALOE, despite a simple algorithm, is much more effective than all these existing baselines. We will also include the full results on ImageNet for these algorithms in the camera-ready version.
>
> We have also added the computational complexity section as Appendix C. In addition, in 5.2.5, we also conducted additional ablation studies by using gaussian mixture models and mini batch K-means as the clustering subprocedure. Both of these are more computationally efficient than simple K-means, and obtains similar performance as the procedure using K-means. This suggests our algorithm is not very sensitive in the choice of the clustering algorithm. In addition, in most active learning algorithms, the selection cost is much less than the cost of training the neural network. In the rare cases where K-means is infeasible, we would recommend practitioners to use alternatives such as GMM and mini batch K-means.

---

> > ### Comment · Reviewer_PXtB · 2024-12-16
> > **Concerns addressed**
> >
> > I want to thank the authors for conducting the additional experiments. My concerns were thoroughly addressed, and I appreciate the effort put into clarifying the points I raised. Moreover, I did not identify any critical issues in the other reviews, and I am satisfied with the current state of the manuscript.

---

### Decision · Action_Editor_sxLt · 2025-01-29

**Recommendation:** Accept as is

**Comment:**

The reviewers initially had concerns about the motivation for the proposed method, the scope of the experiments, and the baselines. All of these concerns have been sufficiently addressed in the rebuttal.

**Audience:**

The reviewers agree that applying active learning to an open-world setting, while not necessarily groundbreaking, is an interesting and useful contribution that can be expected to catch the interest of some part of the TMLR audience.

**Claims And Evidence:**

All reviewers agreed that the claims of the paper are sufficiently supported by empirical evidence. They find that the paper is well-written and the proposed method performs well.